# Opposite Growth Responses of *Alexandrium minutum* and *Alexandrium catenella* to Photoperiods and Temperatures

**DOI:** 10.3390/plants10061056

**Published:** 2021-05-25

**Authors:** Ping Li, Qun Ma, Su Xu, Wenha Liu, Zengling Ma, Guangyan Ni

**Affiliations:** 1Key Laboratory of Marine Biotechnology of Guangdong Province, Shantou University, Shantou 515063, China; liping@stu.edu.cn (P.L.); 14qma@stu.edu.cn (Q.M.); suxu@stu.edu.cn (S.X.); whliu@stu.edu.cn (W.L.); 2Southern Marine Science and Engineering Guangdong Laboratory, Guangzhou 519082, China; 3National and Local Joint Engineering Research Center of Ecological Treatment Technology for Urban Water Pollution, Wenzhou University, Wenzhou 325035, China; mazengling@wzu.edu.cn; 4Key Laboratory of Vegetation Restoration and Management of Degraded Ecosystems, South China Botanical Garden, Chinese Academy of Sciences, Guangzhou 510160, China

**Keywords:** growth, photoperiod, temperature, rubisCO, respiration, antioxidant activity, cell size, *Alexandrium*

## Abstract

Shift of phytoplankton niches from low to high latitudes has altered their experienced light exposure durations and temperatures. To explore this interactive effect, the growth, physiology, and cell compositions of smaller *Alexandrium minutum* and larger *A. catenella*, globally distributed toxic red tide dinoflagellates, were studied under a matrix of photoperiods (light:dark cycles of 8:16, 16:8, and 24:0) and temperatures (18 °C, 22 °C, 25 °C, and 28 °C). Under continuous growth light condition (L:D 24:0), the growth rate (µ) of small *A. minutum* increased from low to medium temperature, then decreased to high temperature, while the µ of large *A. catenella* continuously decreased with increasing temperatures. Shortened photoperiods reduced the µ of *A. minutum*, but enhanced that of *A. catenella*. As temperature increased, cellular Chl *a* content increased in both *A. minutum* and *A. catenella*, while the temperature-induced effect on RubisCO content was limited. Shortened photoperiods enhanced the Chl *a* but reduced RubisCO contents across temperatures. Moreover, shortened photoperiods enhanced photosynthetic capacities of both *A. minutum* and *A. catenella,* i.e., promoting the PSII photochemical quantum yield (F_V_/F_M_, Φ_PSII_), saturation irradiance (E_K_), and maximum relative electron transfer rate (rETRmax). Shortened photoperiods also enhanced dark respiration of *A. minutum* across temperatures, but reduced that of *A. catenella*, as well as the antioxidant activities of both species. Overall, *A. minutum* and *A. catenella* showed differential growth responses to photoperiods across temperatures, probably with cell size.

## 1. Introduction

Global warming has increased the temperature in surface oceans [1,2], extending the low thermal limit of phytoplankton and consequently shifting their niches from low to high latitudes [3,4,5], which has led to the new appearance of temperate species at higher latitudes and even subpolar regions [3]. Such a niches shift varies the temperatures and light exposure durations (i.e., diel light:dark cycle) experienced by phytoplankton. Temperature is well known to regulate phytoplankton growth through affecting their cellular biochemical reactions that are catalyzed by the involved enzymes, such as ribulose-1,5-bisphosphate carboxylase/oxygenase (RubisCO), and thus altering their physiological activities [6]. For example, low temperature reduces the half-saturation constant of RubisCO and lowers the energetic requirement of CO_2_ acquisition [7,8], while high temperature enhances RubisCO activity and gene expression and increases the organic carbon biosynthesis and accumulation [9]. The photoperiod that usually alters the cell-received maximum irradiance or spectral compositions and day or night lengths [10,11,12,13] is another important factor to entrain phytoplankton to a daily rhythm [10,14,15] and to regulate their growth [11,16,17]. In nature, the photoperiod varies with latitudes, seasons, and depths. At high latitudes it can even vary between total darkness and constant illumination from winter to summer. Such a seasonal variation is smaller at low latitudes, but the mixing induced by extreme episodic weathers, e.g., typhoons [18], can abruptly vary the experienced light exposure of cells over a short scale. The light exposure duration often modulates the organic matters accumulation and energy metabolism within cells [19,20]. Previous studies showed that the photoperiod regulates gene expression [21,22,23], cell division [24,25] and growth [11,16,17], and even bloom dynamics of phytoplankton [26,27]. To date, however, only a few studies have examined the interactive effects of temperature and photoperiod upon phytoplankton physiology and metabolism [11,28,29,30,31]. 

Dinoflagellates, the second largest group next to diatoms in species numbers, are important primary producers in the global oceans [32,33]. Cell size of dinoflagellates spans a wide range [34] and strongly influences their growth, photosynthetic activities, and adaptive traits [34,35,36,37,38]. As compared to large cells, small ones usually have more efficient nutrient utilization capacity [39], retain a higher photosynthetic rate under limiting light [17], and are more susceptible to photoinhibition under excessive light [40]. Small cells also have a volume limit for the maximum reserves of energy, e.g., carbohydrates and lipids [37], which is a consequent disadvantage for the dark respiration-fueled synthesis of proteins or structural apparatus [20,41]. Cell size also varies the growth yield that is supported by the coupling of reductant generation to carbon fixation and carbohydrate retention [16,34,42,43], which may cause interspecific differences in growth responses to temperature and photoperiod [11]. 

In this study, we aimed to understand the interacting effects of photoperiod and temperature upon the responses of growth rate and physiology of dinoflagellate *Alexandrium* species with different cell sizes. *Alexandrium* is one of the common dinoflagellates genera and worldwide distributes in all types of marine habitats including eutrophic, mesotrophic, and oligotrophic waters [44,45]. The *Alexandrium* genus contains many harmful species that initiate harmful algal blooms (HABs), cause paralytic shellfish poisoning in both fish and bivalve mollusks, and ultimately harm human health [46,47]. We thus selected two *Alexandrium* strains differing by ~230-fold in cell biovolume, a smaller *Alexandrium minutum* and a larger con-generic *Alexandrium catenella*, and grew them under a matrix of photoperiods and temperatures at an expected optimal instantaneous growth light intensity [17]. We measured the growth, photosynthetic capacity, dark respiration, and antioxidant capacity, as well as cellular chlorophyll *a* (Chl *a*) and RubisCO contents to explore the underlying mechanisms that link the growth rate to cell size, photoperiod, and temperature. Our results would be helpful for understanding about the ecological effects of the changing temperature and light duration upon *Alexandrium* species in a global warming scenario. 

## 2. Results

### 2.1. Growth Rate

In small dinoflagellate *A. minutum*, under continuous 24 h growth light (L:D 24:0) the growth rate (µ) increased from 0.37 ± 0.01 d^−1^ at a temperature of 18 °C to maximal 0.54 ± 0.01 d^−1^ at 25 °C and thereafter decreased to 0.41 ± 0.01 d^−1^ at 28 °C (Figure 1A). Shortened photoperiod lengths (i.e., L:D 8:16 and 16:8) significantly decreased the µ of *A. minutum* across temperatures (*p* < 0.05), and under a L:D cycle of 16:8, the response of µ to growth temperature became weaker. In large *A. catenella*, however, the µ under L:D 24:0 linearly decreased from 0.48 ± 0.02 d^−1^ to 0.19 ± 0.01 d^−1^ as the temperature increased from 18 to 28 °C (R^2^ = 0.98, *p* < 0.01). A shorter photoperiod (i.e., L:D 16:8) significantly enhanced the µ across temperatures (*p* < 0.05), while under L:D 8:16, such enhancement solely occurred at 22 and 25 °C conditions. The response of µ to increasing temperatures was strongly influenced by the photoperiod for large *A. catenella* (*p* < 0.01), but not for small *A. minutum*. In addition, the optimal growth temperature of *A. minutum* was ~25 °C (Figure 1A), while that of *A. catenella* seemed to be lower (i.e., between 22 °C and 18 °C) and varied more with photoperiods (Figure 1B).

### 2.2. Cell Compositions

To explore the mechanisms underlying these contrasting growth responses to temperatures and photoperiods between these two species, the cell biovolume-based biochemical compositions and biological activities were analyzed. Under a L:D cycle of 8:16, Chl *a* content in small *A. minutum* increased from 3.22 ± 0.50 to 5.46 ± 0.22 fg µm^−3^ as the temperature increased from 18 to 28 °C (Figure 2A); extended photoperiods drastically reduced the Chl *a* content (*p* < 0.05), e.g., it reduced by 31% and 70% at 28 °C under L:D 16:8 and L:D 24:0, respectively. In large *A. catenella*, Chl *a* content increased from 0.35 ± 0.02 to 0.48 ± 0.01 fg µm^−3^ with increasing temperatures under shorter L:D 8:16, which was higher than under a longer L:D of 16:8 and 24:0 (*p* < 0.01) (Figure 2B). In contrast to Chl *a*, under the longest L:D, 24:0, the RubisCO contents in both *A. minutum* (0.95 ± 0.07 fg µm^−3^) and *A. catenella* (0.92 ± 0.04 fg µm^−3^) were higher (*p* < 0.05) than that under a shorter L:D of 16:8 and 8:16 across temperatures (Figure 2C,D). In addition, Chl *a* content in *A. minutum* was about 10-fold higher than that in *A. catenella* (Figure 2A,B), while the RubisCO contents were similar in both species (Figure 2C,D).

### 2.3. Chlorophyll Fluorescence

In small *A. minutum*, the maximum photochemical quantum yield of PSII (F_V_/F_M_), an indicator of photosynthetic capacity, was 0.64 ± 0.02 and showed a limited variation across temperatures and photoperiods (Figure 3A). In large *A. catenella*, however, the F_V_/F_M_ under a shorter photoperiod (L:D 16:8) increased from 0.44 ± 0.04 to 0.62 ± 0.02 as the temperature increased from 18 to 28 °C, and an extended photoperiod duration (L:D 24:0) drastically reduced the F_V_/F_M_ by 34% across temperatures (Figure 3B). The extended photoperiod also reduced the effective PSII quantum yield (Φ_PSII_) of both *A. minutum* and *A. catenella*, especially at higher temperatures (Figure 3C,D). The rapid light curve-derived parameters, i.e., light utilization efficiency (*α*), saturation irradiance (E_K_), and maximum relative electron transfer rate (rETR_max_), are shown in Figure 4. The α of *A. minutum* was 0.23 ± 0.03, with a limited effect of temperature and photoperiod (Figure 4A), while that of *A. catenella* seemed to increase with temperature (Figure 4B). Moreover, the E_K_ of *A. minutum* was much lower than that of *A. catenella* across temperatures (*p* < 0.01) and was significantly reduced by extended photoperiods (*p* < 0.05) (Figure 4C,D). Similarly, the rETR_max_ was significantly reduced by extended photoperiods as well (*p* < 0.05) (Figure 4E,F).

### 2.4. Dark Respiration and Antioxidant Activity

In *A. minutum* the cell biovolume-based dark respiration rate (Rd) was higher under shorter photoperiods (*p* < 0.01), while in *A. catenella* the Rd was lower, as compared to longer photoperiods (*p* < 0.01) (Figure 5A,B). The Rd of both species slightly increased with temperature. On the contrary to the Rd, cellular antioxidant activity in both *A. minutum* and *A. catenella* was higher under longer L:D 24:0 than shorter L:D 16:8 or L:D 8:16 across temperatures (*p* < 0.05) (Figure 5C,D). The increased temperature had a limited effect on the antioxidant activity of *A. minutum*, but significantly decreased that of *A. catenella* (R^2^ = 0.61, *p* < 0.01). Finally, the Rd of *A. minutum* was about 1000-fold higher than that of *A. catenella*, while the antioxidant activity was about 10-fold higher.

## 3. Discussion

*Alexandrium minutum* and *A. catenella*, which naturally occur in estuarine, coastal, and oceanic waters [45], represent a wide panel of dinoflagellates in cell size [34] and have demonstrated cell-size dependencies in growth, photosynthesis, and cell compositions, etc. [17,35,36,37]. Growth responses to temperatures and photoperiods by these two *Alexandrium* species with ~230-fold differences in biovolume were opposite, as well as dark respiration responses; moreover, the cellular Chl *a* content of both species was higher under shorter photoperiods and increased with increasing temperatures (Figure 6). Our results indicate that *A. minutum* and *A. catenella* respond differently to photoperiods across temperatures, probably with cell size as well as different strains within species.

The size of a phytoplankton cell usually varies with environmental variables that often interrupt the balance between cell division and photosynthesized organic matter accumulation [39,48]. In this study, no statistical cell-size variation among all combinations of photoperiod and temperature were observed in both *A. minutum* and *A. catenella*, which may be attributed to their mixotrophy, which can diminish cell-size selection if the resources of, e.g., nutrients are insufficient [48] through praying on other organisms or organic detritus [33,49]. To make the data more comparable, the interspecific biochemical and physiological parameters of these two species were normalized to cell biovolume [11,17].

Both *Alexandrium* species showed higher cellular Chl *a* content under shorter photoperiods. Indeed, phytoplankton cells usually accumulate more Chl *a* under insufficient light to optimize their light harvesting capacity [16,17], to support photosynthesis or growth. Apart from Chl *a* content, their light harvesting capacities vary with surrounding light as well [10]. In this study, we used the inverse of Chl *a* (1/Chl *a*) as an indicator of light harvesting capacity per chlorophyll. For *A. minutum* the 1/Chl *a* responded linearly with increasing RubisCO contents among all photoperiod and temperature combinations (R^2^ = 0.54, *p* < 0.05) (Figure 7A), indicating that the stronger Chl *a* light harvesting capacity supports more RubisCO for energy requirements. For *A. catenella*, however, the 1/Chl *a* showed a smooth saturating response to RubisCO (R^2^ = 0.85) (Figure 7B). Moreover, it is well known that the diel L:D cycle regulates cellular gene transcription [23] and organic matter production and accumulation [16,19], leading to a strong diel rhythm in cellular proteins including RubisCO [22], as well as carbohydrates [17]. In this study, cellular RubisCO contents in both *A. minutum* and *A. catenella*, which showed a limited variation across temperatures, were systematically higher under continuous light than that under shorter photoperiods, suggesting the influence of photoperiod on the accumulation of enzymes [20,50]. A higher cellular RubisCO content usually supports a higher growth rate under continuous L:D 24:0 [51]; this is the case for small *A. minutum* but not for large *A. catenella.* This phenomenon also occurred in diatoms *Thalassiosira* species, the large cells of which were found to need a dark period to clear the products of excess light-driven reactive oxygen stress [16].

Maximal PSII photochemical quantum yield (F_V_/F_M_) is usually used to indicate the physiological and photosynthetic status of algae. As the photoperiod extended, the F_V_/F_M_ in small *A. minutum* reduced just at 28 °C, while that in large *A. catenella* reduced in all temperature treatments. Moreover, the F_V_/F_M_ in *A. minutum* was systematically higher and decreased less with extended photoperiods, as compared with *A. catenella*. These patterns probably show the contribution of higher antioxidant activity in smaller species, even though package effects in larger cells can screen their photochemical processes from damage [40,52]. Small *A. minutum* also showed a lower rapid light curve-derived threshold for light saturation of electron transport (E_K_) and a lower maximum relative electron transport rate (rETR_max_). The lower onset of light saturation in *A. minutum* may be attributable to its higher Chl *a* content, which contributes to the increased light harvesting capacity. On the other hand, dark respiration generally provides energy requirements for cellular defense systems, e.g., repairing photoinactivated PSII [19] or maintaining antioxidant protection [17] as well as fueling cell division [53], through consuming photosynthetic products from light periods [20]. Consistently, a high respiration rate and high antioxidant ability are presented in large *A. catenella*. The antioxidant system-induced scavenges of reactive oxygen species (ROS) in chloroplasts can relieve the damage of cellular components of, e.g., proteins and accelerate the synthesis of PSII core protein D1 thereby raising the repair cycle of PSII [31] and leading to the higher photosynthetic capacity. This process is proposed to be energy-costed, indicated by a positive correlation of 1/Chl *a* and antioxidant ability (Figure 7C,D). The ROS also inactivates the key enzymes including RubisCO, resulting in the reduction of carbon fixation [54]. In this case, no direct evidence was observed about the effects of ROS on RubisCO activity; however, the high RubisCO content together with high antioxidant capacity are presented under continuous L:D 24:0 in both *A. minutum* and *A. catenella*, supporting the protective function of cellular enzymatic defense systems. Furthermore, small *A. minutum* had the lowest dark respiration rate under continuous L:D 24:0, under which they may consume less photosynthetic products and retain more for growing, thus showing the highest growth rate. In contrary to small *A. minutum*, large *A. catenella* showed the highest respiration rate under L:D 24:0 with the lowest growth rate. Finally, the antioxidant ability per cell biovolume in *A. minutum* was about 10-fold higher than that of *A. catenella*, indicating a great species-specific difference; and the decrease of antioxidant capacity of *A. catenella* with temperature further suggests that the thermal limit of larger *A. catenella* is lower, as compared with smaller *A. minutum*. At present, there is direct evidence showing that the temperate dinoflagellates species, e.g, *N**octiluca scintillans*, have migrated to higher latitudes and even subpolar regions [3]. For *Alexandrium* species, there is no direct time-series data showing such a migration in local areas of the East China Sea; however, the dominating species shifting from diatoms to dinoflagellates has been examined in the coastal area of the South China Sea [55], as well as more frequent occurrence of *Alexandrium* species blooms [49], which may indirectly indicate the effects of increasing temperatures and possibly photoperiods, in particular in the area of the East China Sea. 

## 4. Materials and Methods

### 4.1. Culture Protocol

The temperate coastal red tide dinoflagellates *Alexandrium minutum* (CCMA-090) (~40 µm^3^ in biovolume) and *Alexandrium catenella* (CCMA-007) (~9300 µm^3^) were obtained from the Xiamen University Center for Marine Phytoplankton and semicontinuously cultured with sterilized f/2 medium [56] in 300 mL glass conical bottles under an expected optimal growth light of 150 µmol photons m^−2^ s^−1^ [17]. These species were originally isolated from the East China Sea and produce paralytic shellfish toxins [57]. During the cultivations, all the bottles were manually shaken 2–3 times a day and arbitrarily distributed in a growth chamber. Light in the chamber was provided with four fluorescent tubes (PHILIPS 18W T8 6500K), and the light intensity was measured with a microspherical quantum sensor (US-SQS, Waltz, Germany) that was submersed in a culture bottle filled with seawater. 

### 4.2. Experimental Design

To determine the interactive effects of temperature and photoperiod duration, both *Alexandrium* species were grown under 18, 22, 25, and 28 °C, in diel light:dark (L:D) cycles of 8:16, 16:8, and 24:0. Different strains within species might also display different responses. After the cultures acclimated to growth conditions, the sample from each treatment was taken at 10:00 a.m. to determine the growth rate, photochemical quantum yield of Photosystem II (PSII), dark respiration and antioxidant activity, and cellular Chl *a* and RubisCO contents as follows.

In this study, three independent cultures were used for each species under each combination of temperature and photoperiod, and a total of 36 separate semi-continuous cultures were performed. The light in the chamber was automatically turned on at 08:00 and maintained continuously (24:0 light:dark (L:D) cycle) or turned off at 12:00 (4:20 L:D cycle), 16:00 (8:16 L:D cycle), or 24:00 (16:8 L:D cycle). These L:D cycles were previously used in analogous studies on differently cell-sized dinoflagellates [17] or diatoms [11,16,19], so selecting them makes the results more comparable. The 24:0 L:D cycle, although not ecophysiologically realistic, was used to compare with common laboratory growth condition.

#### 4.2.1. Growth Rate

To track the growth of small dinoflagellate *A. minutum*, the absorbance of culture was monitored at 680 nm (OD_680_) every morning (10:00 am, 2 h after light on) before and after dilutions with fresh media using an ultraviolet-visible spectrophotometer (SP-752, Shanghai Spectrum Instrument Co., Ltd, Shanghai, China). To track the growth of larger *A. catenella*, duplicate 2 mL cultures were collected at 10:00 a.m. every day, fixed with Lugel’s solution to a 5% final concentration, and the cells were numerated with a Sedgewick Rafter chamber under an inverted microscope (TS2, NIKON, Japan). 

The specific growth rate (µ, d^−1^) was estimated as:µ (d^−1^) = (LN(Nt) − LN(No))/(t − to)(1)
where Nt is the OD_680_ or cell number at time t and N0 is the OD_680_ or cell number at time 0. After at least 12 semicontinuous dilutions with fresh media, the cultures grew through more than 9 cellular generations under each temperature and photoperiod combination.

To estimate cell volume, the fixed cultures were observed under microscopy and two perpendicular cell diameters were measured, one along the cingulum (D_1_) and another along the sagittal plane (D_2_). The cell volume (V) was calculated using an ellipsoid model [58] as: V = 4π/3 × (D_1_/2)^2^ × (D_2_/2)(2)

#### 4.2.2. Maximal PSII Quantum Yield 

To obtain the maximal PS II photochemical quantum yield (F_V_/F_M_), 5 mL culture from each replicate of each treatment was taken and dispensed into a cuvette and dark-adapted for 5 min at each growth temperature. Then, the base line fluorescence (F_O_) and dark-adapted maximal fluorescence (F_M_) were measured with a portable PAM fluorometer (PAM WATERED, Walz, Germany), under a saturating white light pulse of ~5300 µmol photons m^−2^ s^−1^ (800 ms) in the presence of a weak modulated measuring light. After this, instantaneous fluorescence (F_t_) and maximal fluorescence (F_M_’) were measured at growth light to obtain the effective PSII quantum yield (Φ_PSII_). The F_V_/F_M_ and Φ_PSII_ were calculated [59,60] as:F_V_/F_M_ = (F_M_ − F_O_)/F_M_; Φ_PS II_ = (F_M_’ − F_t_)/F_M_’ (3)

At the same time, the rapid light curve (RLC) was measured using the PAM fluorometer under 9 different photosynthetically active photon flux densities (PFD, µmol photons m^−2^ s^−1^), and each of them were applied for 10 s. The relative electron transport rate (rETR) was estimated [59] as:rETR = (F_M_’ − F_t_)/F_M_’ × 0.5 × PAR(4)
where F_M_’ and F_t_ represent the maximal and instantaneous fluorescence under each of 9 PFD levels. The photosynthetic parameters of maximum rETR (rETR_max_), light-utilization efficiency (α), and saturating irradiance (E_K_) were derived from the RCL curves [61] as:rETR = rETRmax × (1 − e − (α × PFD/rETRmax))(5)
EK = rETRmax/α(6)

#### 4.2.3. Dark Respiration

A 2 mL sample from each culture bottle was collected and dark-adapted for 300 s within a chamber with temperature control at growth temperature. The decrease of oxygen concentration was tracked using a liquid oxygen electrode (Hansatech Instrument Ltd, Chlorolab 2, UK). The cell volume-based respiration rate was calculated by dividing the oxygen decline rate by {cells ml^−1^ × cell biovolume}. 

#### 4.2.4. Anti-Oxidant Activity

Fifteen mL culture was centrifuged (7000× *g*) for 5 min; after removing supernatant, the cell pellet was collected, mixed with 3 mL buffer (pH 8.0) containing 20 mM Tris, 1 mM EDTA, 10 mM MgCl, 50 mM NaHCO_3_, and 5 mM β-mercaptoethanol and ultrasonicated for 5 min in an ice bath. The homogenized extract was centrifuged (7000× *g*) again for 10 min, and the total cellular antioxidant activity was measured following the protocol of the antioxidant activity assay kit (A015-2, Nanjing Jiancheng Biological Engineering Company, China) [17]. Finally, the antioxidant ability was transformed as the multiples of Trolox, a vitamin E-like material. 

#### 4.2.5. Cellular Composition

To measure Chl *a* content, duplicate 10 mL culture was centrifuged (7000× *g*) for 5 min to get cell pellets. After discarding the supernatant, 3 mL of 90% acetone (*v*/*v*) saturated with magnesium carbonate was added and extracted overnight at 4 °C in the dark. After centrifugation (7000× *g*, 5 min) again, the absorbance of the supernatant at 664, 630, and 750 nm was measured using the photospectrometer. The Chl *a* was estimated following Jeffrey and Humphrey [62]:[Chl *a*] = 11.47 × (A_664_ − A_750_) − 0.4 × (A_630_ − A_750_)

To measure the RubisCO content, 15 mL culture was collected, centrifuged (7000× *g*, 5 min) to get a cell pellet, and mixed with 5 mL buffer (pH 8.0, see above) and ultrasonicated for 5 min in an ice bath. After centrifugation (7000× *g*, 5 min) again, the supernatant was used to measure RubisCO with an enzyme-linked immunosorbent assay kit (CUSABIO CSB-E16686Pl Quant Kit) [63].

### 4.3. Data Analysis

Data were presented as mean and standard deviations (mean ± SD). Paired t-tests, one-way ANOVA with Bonferroni post-tests (Prism 5, Graphpad Software), and comparisons of linear curve fit were used to detect significant differences among cultures of each diel L:D cycle and temperature combination for each species. The confidence level for statistical tests was set at 0.05.

## 5. Conclusions

Large differences were observed in growth and physiological responses of differently cell-sized *Alexandrium* species to a matrix of photoperiods and temperatures. Small *A. minutum* grew faster under continuous growth light, while large *A. catenella* grew faster under short photoperiods. Shortened photoperiods enhanced cellular Chl *a* content and photosynthetic capacity, but reduced RubisCO content and antioxidant activity. The optimal growth temperature of *A. minutum* was higher than that of *A. catenella*. Our results complement others [3,17] to show that the niches shift may alter *Alexandrium* species compositions in particular in the East China Sea, because different taxa show differential, even opposite, growth responses to photoperiod and temperature changes, as well as different responses of different strains within species.

## Figures and Tables

**Figure 1 plants-10-01056-f001:**
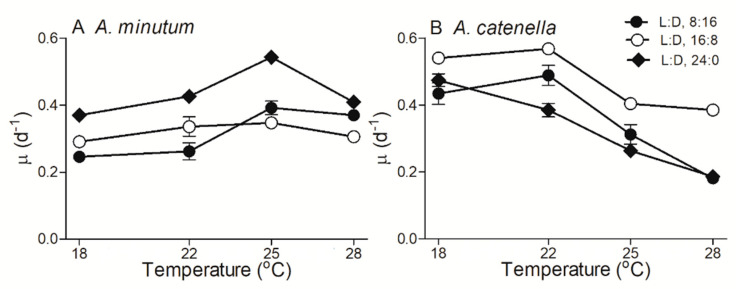
Growth rate (µ, d^−1^) versus culture temperature (°C) for smaller dinoflagellate *Alexandrium minutum* (**A**) and larger *Alexandrium catenella* (**B**) under light:dark (L:D) cycles of 8:16, 16:8, and 24:0 and temperatures of 18, 22, 25, and 28 °C. Points show averages of three growth determinations on independently grown cultures; error bars show standard deviations (n = 3), often within symbols.

**Figure 2 plants-10-01056-f002:**
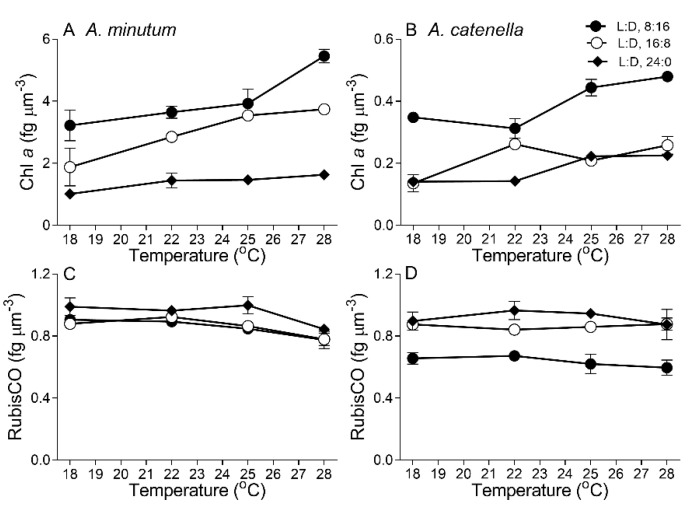
Cell biovolume-based chlorophyll a (**A**,**B**; Chl *a*, fg µm^−3^) and RubisCO contents (**C**,**D**; fg µm^−3^) across growth temperature (°C) for *A. minutum* (**A**,**C**) and *A. catenella* (**B**,**D**) at L:D cycles of 8:16, 16:8, and 24:0. Note: there is a 10-fold difference in the Y-axis scales of panels A and B. Points show averages of three determinations on independently replicated cultures; error bars show standard deviations, often within symbols.

**Figure 3 plants-10-01056-f003:**
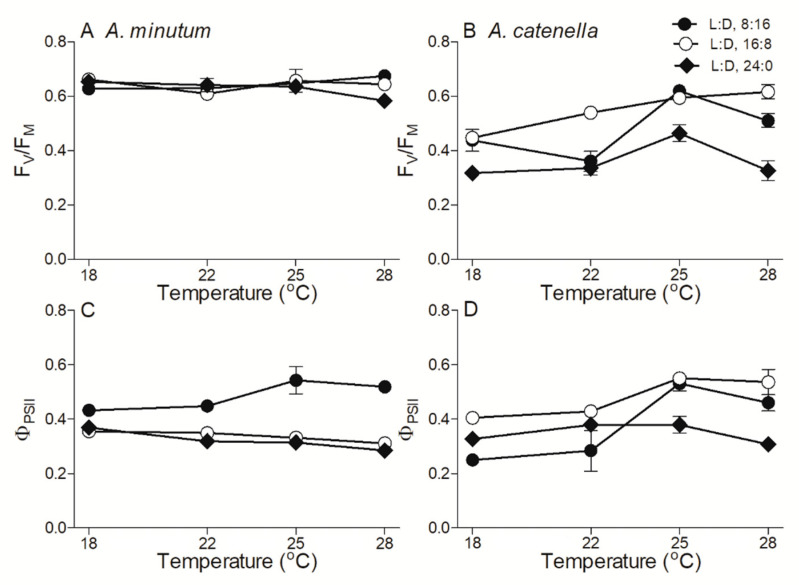
Maximal photochemical quantum yield (F_V_/F_M_) of Photosystem II (PSII) (**A**,**B**) and effective PSII quantum yield (**C**,**D**) across growth temperature (°C) for *A. minutum* (**A**,**C**) and *A. catenella* (**B**,**D**) at L:D cycles of 8:16, 16:8, and 24:0. Points show averages of three determinations on independently replicated cultures; error bars show standard deviations, often within symbols.

**Figure 4 plants-10-01056-f004:**
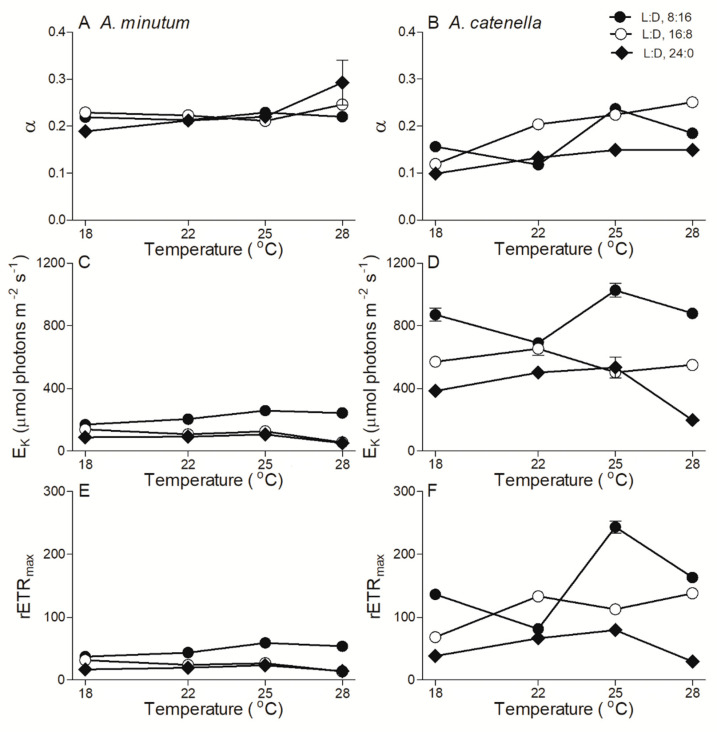
The rapid light curve-derived light utilization efficiency (α) (**A**,**B**), saturation irradiance (E_K_, µmol photons m^−2^ s^−1^) (**C**,**D**), and maximal relative electron transfer rate (rETRmax) (**E**,**F**) across growth temperature (°C) of *A. minutum* (**A**,**C**,**E**) and *A. catenella* (**B**,**D**,**F**) at L:D cycles of 8:16, 16:8, and 24:0. Points show averages of three determinations on independently replicated cultures; error bars show standard deviations, often within symbols.

**Figure 5 plants-10-01056-f005:**
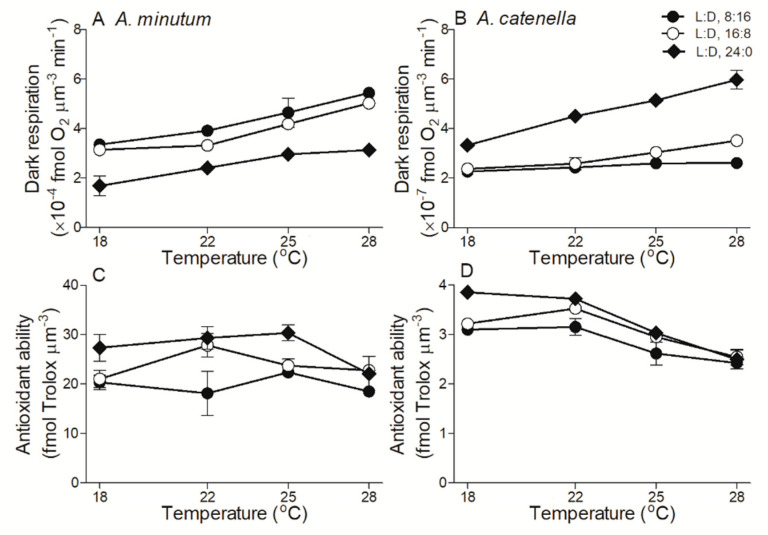
Cell biovolume-based dark respiration (**A**,**B**; fmol O_2_ µm^−3^ min^−1^) and antioxidant capability (**C**,**D**; fmol Tolox µm^-3^) across growth temperature (°C) for *A. minutum* (**A**,**C**) and *A. catenella* (**B**,**D**) at L:D cycles of 8:16, 16:8, and 24:0. Note: there is a 10-fold difference in the Y-axis scales of panels C and D. Points show averages of three determinations on independently replicated cultures; error bars show standard deviations, often within symbols.

**Figure 6 plants-10-01056-f006:**
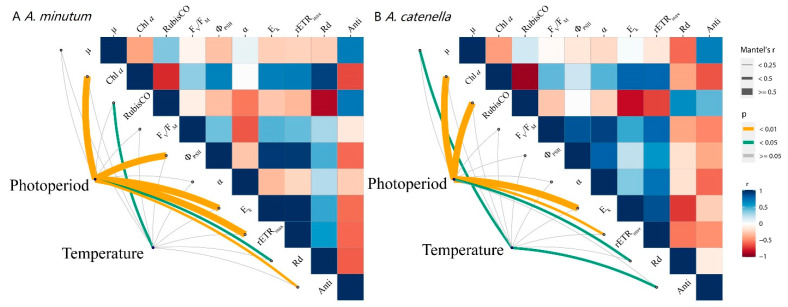
Effects of temperature/photoperiod to biochemical and physiological parameters and their relations for *A. minutum* (**A**) and *A. catenella* (**B**), with color gradient denoting Pearson’s rank correlation coefficients and edge width showing *p* value.

**Figure 7 plants-10-01056-f007:**
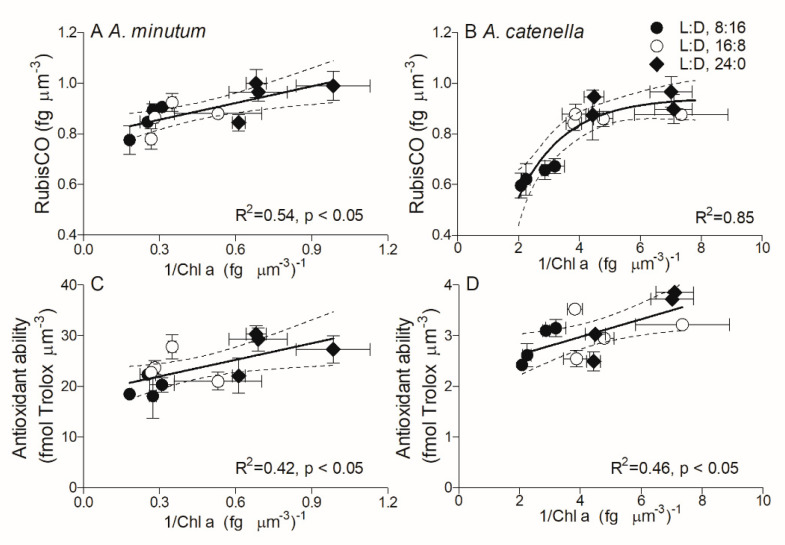
Cell biovolume-based RubisCO content (**A**,**B**, fmol µm^−3^) and antioxidant capability (**C**,**D**, fmol Tolox µm^−3^) versus biovolume-based 1/Chl *a* [(fg µm^−3^)^−1^] for *A. minutum* (**A**,**C**) and *A. catenella* (**B**,**D**) at L:D cycles of 8:16, 16:8, and 24:0. Bold lines in panels (**A**,**C**,**D**) show pooled linear regression (R^2^ = 0.42–0.54, *p* < 0.05) and in panel B shows one phase decay (R^2^ = 0.85); thin dashed lines show 95% confidence intervals on the fitted curve.

## Data Availability

The data that support the findings of this study are available from the corresponding author upon reasonable request.

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
