# Peer review of "Opposite Growth Responses of Alexandrium minutum and Alexandrium catenella to Photoperiods and Temperatures"

_plants, 2021, doi:10.3390/plants10061056_

Round 1
Reviewer 1 Report
Review for the paper : Opposite Growth Responses of Alexandrium minutum and Alexandrium catenella to Photoperiods and Temperatures
General comments:
In this manuscript, the authors selected two Alexandrium strains differing in cell biovolume, a smaller Alexandrium minutum and a larger A. catenella, and grew them under a matrix of photoperiods and temperatures. They measured the growth, photosynthetic capacity, dark respiration and antioxidant capacity, cellular chlorophyll a and RubisCO content of selected dinoflagellate species to investigate the relationship between growth and different photoperiods under different temperatures.
Although the topic of the manuscript is very interesting, there are still some points that need to be clarified, especially in the discussion section. There is a lot of repetition of results in the discussion, and more attention should be paid to what these results mean in a particular context. Also, it is very difficult to follow the text because of the repetitive figures in the discussion. It is not readable. Parentheses should be kept only for necessary repetition of results, and parentheses with figure citations should be deleted completely.
I would suggest the authors to make a schematic diagram of the experimental setup. Also, the results section should include subheadings (e.g., either starting from different photoperiods or different temperatures), and finally, the authors should summarize the main differences between A. minutum and A. catenella in a simple table to visualize the main results of all these measurements.
Specific comments:
Line 35 rephase- „relaxing“
Line 52 rephase… „second to diatoms“
Line 84. what is the duration of a particular photoperiod?
Figure 1. For clarity, only the temperatures used in the study should be indicated in the diagrams.
Line 104: What does "prolonged photoperiod" mean?
Figure 2. scales for chl a should be as uniform as possible for better comparison.
Figure 4. uniform scales! Whether all the components presented are equally important to the subject --- light-utilization efficiency, saturation irradiance, relative electron transfer rate? Could the authors present the most relevant measurement corresponding to light harvesting?
Line 260: optical density?
Line 305: Is 300 s enough for Alexandrium to adapt to dark?
Why did the authors use inverted 1/chla a for Rubisco activity...? does the content of chl a indicate the proportional ability of the cell to accumulate new organic matter and then consequently increase biomass? Reference?
Discussion
Line 171: "In this study, we detected no statistical cell size changes among all combinations of photoperiod and temperature in both A. minutum and A. catenella."---- Please rephrase and explain this sentence.
Lines 181-182: Is there any reference fo inverse Chla and light harvesting?
Lines 188: repetitively, explain what this difference indicates.
Line 194 please explain...."suggesting the influence of photoperiod on the accumulations of enzymes or structural materials"
Line 191: There are contradictory statements in the following two sentences, could the authors please explain what they mean...." cellular RubisCO contents in both A. minutum and A. catenella... were systematically higher under continuous lights than that under shorter photoperiods..." " Higher cellular RubisCO content usually supports a higher growth rate under continuous L:D 24:0 [47]; this is the case for small A. minutum but not for large A. catenella..."
Line 199: what about small cells? How do they purify excess light-driven reactive oxygen stress products?
Lines 260-265: I don't understand why the authors measured growth rate in a different way for A. minutum than for A. catenella. Why didn't they use the same approach (e.g. Sedgwick Rafter chamber). Is it comparable; optical density and abundance?
Conclusion section
Lines 345-350: I understand that the authors wanted to compare their results with the previous ones, but this part should go in the discussion, and the sentence should be rephrased.
Reviewer 2 Report
See attached file

Reviewer 3 Report
Opposite Growth Responses of Alexandrium minutum and Alexandrium catenella to Photoperiods and Temperatures
The aim of the study was to observe differences in the response of Alexandrium species to an increase in temperature depending on the cell-size and photoperiod. The research was conducted on two Alexandrium species, A.minutum and A. catenella, which is about 230-fold larger in cell biovolume. The research is interesting because of the small number of studies dealing with the interactive effects of temperature and photoperiod upon phytoplankton growth, physiology and metabolism. The results can be applied in a possible prediction of the impact of climate change and global warming on the marine ecosystem.
Author Response
Thank you very much for the positive comments.
Reviewer 4 Report
The authors describe the physiological response of two Alexandrium strains to changing temperature and photoperiod. In the light of our changing oceans this information is highly topical. The manuscript and figures are well presented and the results sound, but there are a few caveats. Do the authors expect all Alexandrium strains within these Alexandrium species to behave the same? The Alexandrium species complex contains various species and strains isolated from different geographical regions across the world and care should be taken to not overstate the significance of the results in terms of species comparison based on single strains within each species.
The results are likely most representative for the area where the two strains have been isolated. It would be nice to tie this information back into the discussion to interpret how the results obtained here for the two Alex strains are likely to change their abundance in the future in the waters that they were isolated from. How are conditions expected to change in the area where these two strains occur? Have these strains of Alexandrium already extended their geographical range in the waters from where they have been isolated?
Methods:
Was the light supplied to the cultures ramped throughout the light period to simulate dusk/dawn conditions or simply provided as on/off?
Please also provide details from where and when these two strains were isolated. Have the strains been tested for paralytic shellfish toxin production? Do the authors know which Group of Alexandrium catenella this strain belongs to?
Results:
Please add the standard deviations to the average values given in the results section.
Specific comments
L20-21: suggest change of wording to decreased with increasing temperatures?
L55-57: is this hold true? Perhaps consider revising the sentence. A quick google revealed several works directly focusing on photoperiod and temperature effects on phytoplankton:
Shatwell, T., Köhler, J. and Nicklisch, A., 2014. Temperature and photoperiod interactions with phosphorus-limited growth and competition of two diatoms. PLoS One, 9(7), p.e102367.
Shatwell, T., Köhler, J. and Nicklisch, A., 2013. Temperature and photoperiod interactions with silicon-limited growth and competition of two diatoms. Journal of plankton research, 35(5), pp.957-971.
Shatwell, T., Nicklisch, A. and Köhler, J., 2012. Temperature and photoperiod effects on phytoplankton growing under simulated mixed layer light fluctuations. Limnology and Oceanography, 57(2), pp.541-553.
Nicklisch, A., Shatwell, T. and Köhler, J., 2008. Analysis and modelling of the interactive effects of temperature and light on phytoplankton growth and relevance for the spring bloom. Journal of Plankton Research, 30(1), pp.75-91.
L95: change lowered to lower
L:105 & 108: should the units be fg per µm, i.e. µm-3 with the minus? Please also add in standard deviations
L:117&119: please add in standard deviations
L168-170: Please note that different strains within species might also display different responses. A note should be inserted here to that effect.
L201: it would be nice to include r^2 on the figure
L349-351: again, different strains within species might also display different responses
Round 2
Reviewer 2 Report
I have reviewed the revised version of manuscript plants-1189403 by Li et al. This study is very similar to the previous study of Xu et al. (2020). The difference is that Xu et al. (2020) studied two species of Thalassiosira (diatom), and Li et al. studied two species of Alexandrium (dinoflagellate). The concluding sentence in the abstract of the Li et al. manuscript is “Our results indicate that growth responses of Alexandrium species to photoperiods across temperatures may vary with species, and possibly with cell size.” The last sentence in the abstract of the Xu et al. (2020) article includes the words, “Our results demonstrated that responses of diatoms Thalassiosira across photoperiods and temperatures vary with species and possibly with cell size.”
These are very similar conclusions and very similar studies, one with a large and small diatom with cell volumes that varied by a factor of 7500, and the other with a small and large dinoflagellate with cell volumes that differed by a factor of 230.
We have known for decades that there are significant differences in the physiology of different species of phytoplankton. That understanding goes back to Hutchinson’s Paradox of the Plankton and has continued up to the present time. With the recent concern about the impacts of climate change, numerous studies have been carried out to determine what the likely impacts will be of the anticipated changes in environmental conditions. Those studies have shown that the responses of marine organisms, including phytoplankton, to the likely effects of climate change are by no means uniform. One noteworthy paper is “Marine Calcifiers Exhibit Mixed Responses to CO2-induced Ocean Acidification” by Ries et al. (2009) that was published in Geology 37: 1131–1134. That paper has been cited 798 times according to the Web of Science.
One point I want to stress in this review is that the idea that not all phytoplankton respond in the same way to changes of environmental conditions is just not news, and that includes the likely changes of environmental conditions associated with climate change. So I am going to disagree with the other reviewers that the discovery that two species of Alexandrium did not respond in the same way to changes of temperature and photoperiod is newsworthy. I do not think it is newsworthy at all.
What I think might be newsworthy would be a demonstration that at least some of the differences could be attributed with confidence to differences in the sizes of the organisms. We already know that different species respond in different ways to environmental changes. What would be helpful is an understanding of why they respond differently. With that kind of understanding, we could make predictions that could enhance our ability to anticipate likely climate change effects. As things stand, all we have from this manuscript is a demonstration that two species do not respond in the same way and a suggestion that the differences may have something to do with cell size. I see the same problem with the Xu et al. (2020) paper.
There are several sentences in the Introduction of the Li et al. paper about differences between small and large phytoplankton cells. What is missing, however, is any follow-up. Based on what we know about cell size effects, how would those differences be expected to affect the responses of cells of different sizes to changes of photoperiod and temperature? That question is never asked by Li et al. Instead, they perform various measurements and then conclude that there are differences between the two species. I just do not find that result to be newsworthy. We have known that there are differences in the responses of different species to changes of environmental conditions for decades, and the 2009 paper by Ries et al. is but one of many examples of how different species respond differently to the changes likely to result from the buildup of CO2 in the atmosphere and ocean. A big problem with this manuscript is that Li et al. never hypothesize how they think differences in cell size might be expected to cause differences in the responses of phytoplankton to changes of photoperiod and temperature, and as a result the experiments they carried out test only the hypothesis that there are differences between species. That hypothesis has been tested over and over again for decades, and we know the answer. What would be very helpful now is some basic understanding of why different species respond differently, and if cell size has something to do with those differences, that would be very useful to know, but this paper provides no clues because it does not test relevant hypotheses.
Author Response
I have reviewed the revised version of manuscript plants-1189403 by Li et al. This study is very similar to the previous study of Xu et al. (2020). The difference is that Xu et al. (2020) studied two species of Thalassiosira (diatom), and Li et al. studied two species of Alexandrium (dinoflagellate). The concluding sentence in the abstract of the Li et al. manuscript is “Our results indicate that growth responses of Alexandrium species to photoperiods across temperatures may vary with species, and possibly with cell size.” The last sentence in the abstract of the Xu et al. (2020) article includes the words, “Our results demonstrated that responses of diatoms Thalassiosira across photoperiods and temperatures vary with species and possibly with cell size.”
These are very similar conclusions and very similar studies, one with a large and small diatom with cell volumes that varied by a factor of 7500, and the other with a small and large dinoflagellate with cell volumes that differed by a factor of 230.
Thank you very much for critical comments on our manuscript.
Yes, we have read the paper of Xu et al. (2020) too, and cited it. We found the dinoflagellates Alexandrium, similar as diatoms Thalassiosira, showed differential physiological responses to growth temperatures and photoperiods, related to different species or cell size. Xu et al. 2020 studied Thalassiosira while we studied Alexandrium, one is for diatom and one is for dinoflagellate. We think it’s normal that different papers give a similar conclusion, and it’s meaningless to argue about this point.
We have known for decades that there are significant differences in the physiology of different species of phytoplankton. That understanding goes back to Hutchinson’s Paradox of the Plankton and has continued up to the present time. With the recent concern about the impacts of climate change, numerous studies have been carried out to determine what the likely impacts will be of the anticipated changes in environmental conditions. Those studies have shown that the responses of marine organisms, including phytoplankton, to the likely effects of climate change are by no means uniform. One noteworthy paper is “Marine Calcifiers Exhibit Mixed Responses to CO2-induced Ocean Acidification” by Ries et al. (2009) that was published in Geology 37: 1131–1134. That paper has been cited 798 times according to the Web of Science. One point I want to stress in this review is that the idea that not all phytoplankton respond in the same way to changes of environmental conditions is just not news, and that includes the likely changes of environmental conditions associated with climate change. So I am going to disagree with the other reviewers that the discovery that two species of Alexandrium did not respond in the same way to changes of temperature and photoperiod is newsworthy. I do not think it is newsworthy at all.
No response I can give, except respect to other reviewers’ comments.
We appreciate the work of Ries et al. (2009) that reported a broad taxonomic range (crustacea, cnidaria, echinoidea, rhodophyta, chlorophyta, gastropoda, bivalvia, annelida) to environmental changes. We used differential cell-sized species in same genus, just considering the problem of comparability.
What I think might be newsworthy would be a demonstration that at least some of the differences could be attributed with confidence to differences in the sizes of the organisms. We already know
that different species respond in different ways to environmental changes. What would be helpful is an understanding of why they respond differently. With that kind of understanding, we could make predictions that could enhance our ability to anticipate likely climate change effects. As things stand, all we have from this manuscript is a demonstration that two species do not respond in the same way and a suggestion that the differences may have something to do with cell size. I see the same problem with the Xu et al. (2020) paper.
We completely agree with the reviewer that some further studies need to be performed for the mechanisms of differential responses of cell-size-related phytoplankton. The transromantic analysis is ongoing now.
There are several sentences in the Introduction of the Li et al. paper about differences between small and large phytoplankton cells. What is missing, however, is any follow-up. Based on what we know about cell size effects, how would those differences be expected to affect the responses of cells of different sizes to changes of photoperiod and temperature? That question is never asked by Li et al. Instead, they perform various measurements and then conclude that there are differences between the two species. I just do not find that result to be newsworthy. We have known that there are differences in the responses of different species to changes of environmental conditions for decades, and the 2009 paper by Ries et al. is but one of many examples of how different species respond differently to the changes likely to result from the buildup of CO2 in the atmosphere and ocean. A big problem with this manuscript is that Li et al. never hypothesize how they think differences in cell size might be expected to cause differences in the responses of phytoplankton to changes of photoperiod and temperature, and as a result the experiments they carried out test only the hypothesis that there are differences between species. That hypothesis has been tested over and over again for decades, and we know the answer. What would be very helpful now is some basic understanding of why different species respond differently, and if cell size has something to do with those differences, that would be very useful to know, but this paper provides no clues because it does not test relevant hypotheses.
We agree with reviewer the answer about different species respond differently to environmental changes is clear. We think Ries et al. (2009) knew this answer too, but they continued performing experiments to test it, published and was accepted extensively. Similarly, in this manuscript we provided the data to support the known answer that differently cell-sized phytoplankton responded differently to temperature and photoperiod changes.